# Neural McKean-Vlasov Processes:
# Inferring Distributional Dependence

## Abstract

McKean-Vlasov stochastic differential equations (MV-SDEs) provide a mathematical description of the behavior of an infinite number of interacting particles by imposing a dependence on the particle density. These processes differ from standard Itô-SDEs to the extent that MV-SDEs include distributional information in their individual particle parameterization. As such, we study the influence of explicitly including distributional information in the parameterization of the SDE. We first propose a series of semi-parametric methods for representing MV-SDEs, and then propose corresponding estimators for inferring parameters from data based on the underlying properties of the MV-SDE. By analyzing the properties of the different architectures and estimators, we consider their relationship to standard Itô-SDEs and consider their applicability in relevant machine learning problems. We empirically compare the performance of the different architectures on a series of real and synthetic datasets for time series and probabilistic modeling. The results suggest that including the distributional dependence in MV-SDEs is an effective modeling framework for temporal data under an exchangeability assumption while maintaining strong performance for standard Itô-SDE problems due to the richer class of probability flows associated with MV-SDEs.

## 1 Introduction

Stochastic differential equations (SDEs) model the evolution of a stochastic process through two functions known as the *drift* and *diffusion* functions. Beginning with Itô-SDEs, where individual sample paths are assumed to be independent, neural representations of the drift and diffusion have achieved high performance in many applications, such as time series and generative modeling [Song et al., 2020, Tashiro et al., 2021].

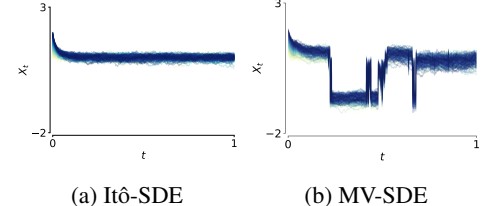

(a) Itô-SDE      (b) MV-SDE

Figure 1: SDE sample paths of a double-well potential, where the particles (a) do not interact and (b) exhibit complex phase transitions as a result only of interaction via weak attraction.

On the other hand, interacting particle systems are also used to model stochastic processes using many of the same characteristics as an Itô-SDE, but they additionally dictate an interaction between the different sample paths [Liggett, 1997]. When the number of particles approaches infinity, these processes generalize Itô-SDEs to *nonlinear SDEs* known as McKean-Vlasov SDEs (MV-SDEs). The nonlinearity arises from the individual particle dependence on the whole particle density, often in the form of a *mean-field* term represented by an expectation with respect to the particle density. This distributional dependence allows for greater flexibility in the time marginal distributions that the MV-SDE can represent versus the Itô-SDE. An

example of the differences between the two frameworks is illustrated in Figure 1 where Figure 1a depicts an Itô-SDE where the sample paths are independent and Figure 1b depicts a MV-SDE where the sample paths interact through distributional dependence. While these models appear in a variety of disciplines such as in finance [Feinstein and Søjmark, 2021], biology [Keller and Segel, 1971], and social sciences [Carrillo et al., 2020], relatively few works have considered the problem of estimating parameters from observations or their application in machine learning tasks.

This brings us to a motivating question:

*(Q1) Can we develop theoretically justified neural architectures to represent MV-SDEs?*

To answer *(Q1)*, we use the relationship between the approximation capabilities of neural networks and properties of MV-SDEs. We consider two ideas: (i) expressing a layer in a neural network as an expectation with respect to a density and (ii) using generative models to capture distributions and generate samples.

Our second question relates the theoretical generality of MV-SDEs to Itô-SDEs:

*(Q2) Does including explicit distributional dependence empirically affect modeling capabilities?*

We discuss a few theoretical properties that motivate this question and answer the question empirically by comparing different architectures for applications in time series and in probabilistic modeling.

## 1.1 Related work

Methods that estimate MV-SDEs from observations often assume known interaction kernels and drift parameters. They then rely on a large number of samples at regularly spaced time intervals to empirically approximate the expectation in the mean-field term [Messenger and Bortz, 2022, Della Maestra and Hoffmann, 2022, Yao et al., 2022, Della Maestra and Hoffmann, 2023]. In Pavliotis and Zanoni [2022], the authors describe a method of moments estimator for the parameters of the MV-SDE. Other approaches concerned analyzing the partial differential equation (PDE) associated with MV-SDEs as in Gomes et al. [2019]. In our work, we are primarily concerned with inference in regions where we have limited time-marginal data and the number of samples is not large. Other applications of MV-SDEs in machine learning topics include estimating optimal trajectories in scRNA-Seq data [Chizat et al., 2022] and stochastic control problems relating to mean-field games [Han et al., 2022]. Ruthotto et al. [2020] considered a machine learning approach for solving certain kinds of mean field games and mean field control problems. Inverse problems can also be solved by deriving an appropriate MV-SDE as the authors describe in Crucinio et al. [2022]. Extensive analysis of the dynamics of the parameters of a neural network under stochastic gradient descent has been conducted using the theory from MV-SDEs, e.g. [Hu et al., 2021]. These methods use a pre-described form of the drift to conduct their analyses whereas we're interested in learning a representation of the drift.

**Our Contributions** To address the lack of non-parametric MV-SDE estimators in the existing literature, this paper contributes the following: First, we present two neural architectures for representing MV-SDEs based on learned measures and generative networks; then, we present three estimators, based on maximum likelihood, used in conjunction with the architectures without prior knowledge on the structure of the drift; next, we characterize the properties of implicit regularization and richer probability flows of these architectures; finally, we empirically demonstrate the applicability of the architectures on time series and generative modeling.

## 2 Properties of MV-SDEs

We begin by describing the background and properties of the transition densities of MV-SDEs. Figure 2 illustrates some of these concepts qualitatively where we first consider non-local dynamics and then consider jumps in the sample paths.

### 2.1 Background

Consider a domain $\mathcal{D} \subset \mathbb{R}^d$ and let $\mathcal{P}_k(\mathcal{D})$ be the space of all probability distributions supported on $\mathcal{D}$ with finite $k$th moment. Let $W_t \in \mathbb{R}^d$ be a $d$-dimensional Wiener process and let $X_t \in \mathbb{R}^d$ be a

solution to the following MV-SDE

$$\mathrm{d}X_t = b(X_t, p_t, t)\mathrm{d}t + \sqrt{\Sigma(X_t, p_t, t)}\mathrm{d}W_t \tag{1}$$

where $p_t$ denotes the law of $X_t$ at time $t$ and $\sqrt{\Sigma}$ denotes the Cholesky decomposition of $\Sigma$. We assume that the *drift* vector $b : \mathbb{R}^d \times \mathcal{P}_2(\mathcal{D}) \times \mathbb{R}_+ \to \mathbb{R}^d$ and the *diffusion* matrix $\Sigma : \mathbb{R}^d \times \mathcal{P}_2(\mathcal{D}) \times \mathbb{R}_+ \to \mathrm{SPD}(\mathbb{R}^{d \times d})$ are globally Lipshitz for the existence and uniqueness of the solution, with $\mathrm{SPD}$ denoting the space of symmetric, positive definite matrices.

We focus on the case where the diffusion coefficient is a known constant, $\sigma$, and focus on estimating the drift, $b$, from data. In addition, for simplicity in analysis, we suppose that $b$ factors linearly into a non-interacting component, and an interacting component, where the mean-field term with dependence on $p_t$ is often written in terms of an expectation, specifically

$$\mathrm{d}X_t = f(X_t, t)\mathrm{d}t + \mathbb{E}_{y_t \sim p_t}[\varphi(X_t, y_t)]\mathrm{d}t + \sigma\mathrm{d}W_t \tag{2}$$

where $f : \mathbb{R}^d \times \mathbb{R}_+ \to \mathbb{R}^d$ can be seen as the Itô drift, the expectation as the mean-field drift, and $\varphi : \mathbb{R}^d \times \mathbb{R}^d \to \mathbb{R}^k$ as the *interaction* function describing the interaction between particles, e.g. attraction with $\varphi(x, y) = -(x - y)$ in Figure 1b and the left side of Figure 2. We also assume that all coefficients exhibit sufficient regularity such that the empirical law converges to the true law of the system (i.e. $\frac{1}{N}\sum_{i=1}^{N}\delta_{X^{(i)}_t} \to_{N\to\infty} p_t(X_t)$), i.e. propagation of chaos holds [Méléard, 1996]. As mentioned, unlike Itô-SDEs which only consider dependence on $X_t$ and $t$, MV-SDEs also depend on the marginal time distribution $p_t$. By introducing a dependence on the marginal law, the transition density of the process satisfies a richer class of functions.

## 2.2 Non-locality of the transition density

Following the background, we describe a favorable property of the MV-SDE that induces non-local dependencies in the state space. The transition density of (2) can be written as the non-linear PDE

$$\partial_t p_t(x) = -\nabla \cdot \left( \underbrace{p_t f(x)}_{\text{Itô Drift}} + \underbrace{p_t \int \varphi(x - y_t)p_t(y_t)\mathrm{d}y_t}_{\text{Non-Local Interactions}} - \underbrace{\frac{\sigma^2}{2}\nabla p_t}_{\text{Diffusion}} \right). \tag{3}$$

This non-local behavior has a variety of implications. For example, the distribution of particles "far away" from a reference particle can affect the behavior of the reference particle. This property is illustrated in the left side of Figure 2 with an example from the mean-field FitzHugh-Nagumo model used to model spikes in neuron activation, leading to interactions between distinct spikes [Crevat et al., 2019]. Notably, this is not possible when considering only the Itô drift, since that operator acts locally on the density.

## 2.3 Discontinuous sample paths

The richer class of densities modeled by MV-SDEs has direct influence on individual sample paths. In a modeling scenario, we may wish to approximate a process that exhibits jumps. For example, in finance, a number of related entities may have common exposure and experience failure simultaneously [Nadtochiy and Shkolnikov, 2019, Feinstein and Søjmark, 2021]. Similarly, in neuroscience, a number of neurons spiking simultaneously results in discontinuities in the sample paths [Carrillo et al., 2013]. The fact

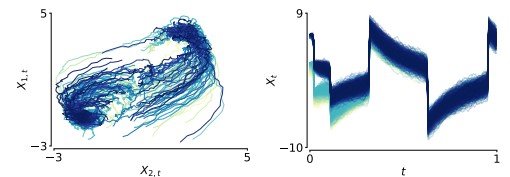

Figure 2: MV-SDE sample paths with non-local dynamics (left) and discontinuities (right).

that the interaction of many particles can cause blowups leads to a remarkable property of MV-SDEs that allows discontinuous paths. The major benefit of this property is that we do not need to consider an additional jump noise process – we only need to specify a particular interaction between the particles to induce the jump behavior. A simple proof for the case of positive feedback is given in Hambly et al. [2019, Theorem 1.1].

Having described the theoretical advantages of MV-SDEs as compared to Itô-SDEs, we will proceed to discuss the neural architectures for representing these processes.

| Implicit Measure (IM) | Empirical Measure (EM) | Marginal Law (ML) |
|---|---|---|
| $\frac{1}{n_w}\sum_{i=1}^{n_w}\varphi(\,\cdot\,,W_0^{(i)})\frac{\mathrm{d}\mathbb{P}_t}{\mathrm{d}\mathbb{P}_0}$ | $\frac{1}{N}\sum_{i=1}^{N}\varphi(\,\cdot\,,X_t^{(i)})$ | $\frac{1}{n_p}\sum_{i=1}^{n_p}\varphi(\,\cdot\,,\hat{X}_t^{(i)})$ |
| | $\left\{X_t^{(i)}\right\}_{i=1}^{N}\sim P_t$ | $\left\{\hat{X}_t^{(i)}\right\}_{i=1}^{n_p}\sim \hat{P}_t$ |

Implicit $p_t$ ← ————————————————————————— → Explicit $p_t$

Figure 3: Schematic comparing neural architectures for modeling MV-SDEs. Implicit measure (IM) architecture uses a mean-field layer that represents particles as learned weights; the empirical measure (EM) architecture computes the expectation with the observed particles; the marginal law (ML) estimates the particle density, computing the expectation with samples from the estimated density.

## 3 Mean-field Architectures

We now describe methods for representing the mean-field drift of a MV-SDE in (2). We first consider a modification of the cylindrical architecture [Pham and Warin, 2022] that empirically computes the expectation using observations, and denote it as the empirical measure (EM) architecture. We then propose two architectures – an architecture based on representing a learned measure with neural weights, denoted as the implicit measure (IM) architecture, and a generative architecture based on representing the marginal law of the samples (ML). Figure 3 provides a schematic of the different architectures and mean-field representations. We denote a function $f$ parameterized by parameters $\theta$ as $f(\cdot;\theta)$.

### 3.1 Empirical measure architecture

Suppose we observe $N$ particles at each time $t$ given by $\{X_t^{(i)}\}_{i=1}^{N}$ and denote the discrete measure associated with these observations as $p_t^\delta = \frac{1}{N}\sum_{i=1}^{N}\delta_{X_t^{(i)}}$. Then, we can use $p_t^\delta$ to approximate the expectation in (2) as

$$\mathbb{E}_{y_t\sim p_t}\left[\varphi\left(X_t,y_t\right)\right] \approx \mathbb{E}_{y_t\sim p_t^\delta}\left[\varphi\left(X_t,y_t;\theta\right)\right] = \frac{1}{N}\sum_{i=1}^{N}\varphi\left(X_t,X_t^{(i)};\theta\right) \tag{4}$$

for a neural network $\varphi(\cdot,\cdot;\theta)$ describing the interaction function between the particles [Pham and Warin, 2022]. Suppose the non-mean field component $f$ is also represented with a neural network $f(\cdot,t;\theta)$. Assuming that $\varphi$ and $f$ are well learned, this architecture can represent the true MV-SDE drift in the limit as the number of observations $N \to \infty$. We refer to this architecture as the *empirical measure* (EM) architecture since at each time step the expectation is taken with respect to the empirical measure derived from the observations.

### 3.2 Implicit measure architecture

While the EM architecture in (4) explicitly defines the relationship between the law $p_t$ and the interaction $\varphi$, it relies on obtaining the empirical measure at each time point. This may be difficult in practice for a variety of reasons such as having few samples or the lack of data at some time points.

Instead, let us first recall that a single layer in a multilayer perceptron (MLP) can be written in terms of an expectation as

$$\mathrm{MLP}^{W,b}(x) = \int \sigma\left(Wx + b\right)\mathrm{d}\nu\left(W,b\right) \tag{5}$$

where the expectation is taken over $\nu(\cdot)$, a measure over the space of parameters $y = (W,b)$, and $\sigma$ is an activation function.

When $\nu = \frac{1}{N}\sum_{i=1}^{N}\delta_{y^{(i)}}$, a discrete measure with $N$ particles, the expectation is exactly a single layer of width $N$, suggesting a correspondence between an empirical measure with $N$ samples and a single layer of width $N$. Building on this correspondence, we propose a mean-field layer:

**Definition 3.1** (Mean-field Layer)**.** Define the weight of the mean-field layer with width $n$ as the matrix $W_0 \in \mathbb{R}^{n\times d}$ and denote its $i$th row as $W_0^{(i)}$. The mean-field layer then is defined by the

operation

$$\mathrm{MF}_{(n)}(\varphi(X_t)) := \frac{1}{n} \sum_{i=1}^{n} \varphi(X_t, W_0^{(i)}) \frac{\mathrm{d}\mathbb{P}_t}{\mathrm{d}\mathbb{P}_0}. \tag{6}$$

The mean-field layer (MF) can be thought of as another layer within the network architecture that approximates the law $p_t$. Each row $W_0^{(i)}$ is of size $R^d$, corresponding to the dimensions of $X_t^{(i)} \in \mathbb{R}^d$. The activation function of the mean-field layer is the average over the augmented dimension over which MF operates. The change of measure $\frac{\mathrm{d}\mathbb{P}_t}{\mathrm{d}\mathbb{P}_0}$ can be learned as part of the estimator of the interaction function, $\varphi(\cdot, \cdot, t; \theta)$. Importantly, the above representation allows modeling mean-field interactions without the need for a full set of observations at each time point and without the need to explicitly represent the distribution $p_t$ at each time point. Assuming that $\varphi$ and MF are well learned, this architecture can represent the true MV-SDE drift in the limit as the width $n \to \infty$. We note empirically that a finite $n$ is sufficient and we provide examples of ablations in the appendix.

A similar analysis can be made for the standard MLP architecture. However, the explicit separation of $f$ and $\varphi$ is not enforced in this case. This leads us to the following remark:

**Remark 3.2** (Itô-SDEs with drift represented using MLPs can model MV-SDEs). *From the above discussion, the expectation with respect to the law $p_t$ may be implicitly represented by a MLP.*

Our motivation is then concerned with how a relatively more explicit distribution dependence with $\varphi$ and MF affect modeling capabilities. This explicit structure lends to an implicit regularization that promotes a smaller norm of the mean-field component under a maximum likelihood estimation framework, which we detail later in Section 5.1.

### 3.3 Marginal law architecture

A solution to the MV-SDE is the pair $(X, p)$ such that $p_t = \mathrm{Law}(X_t)$. In addition, if $p$ is a solution to the SDE in (2), it is also a weak solution to the PDE in (3), and the converse holds. For this reason, $p$ is often itself the main object of study. In the marginal law (ML) architecture, in conjunction with the drift, we introduce a generative model for representing the time-varying density. In this case, we approximate the expectation in (2) as

$$\mathbb{E}_{y_t \sim p_t}[\varphi(X_t, y_t)] \approx \mathbb{E}_{y_t \sim \hat{P}_t}[\varphi(X_t, y_t; \theta)] = \frac{1}{n} \sum_{i=1}^{n} \varphi\left(X_t, \hat{X}_t^{(i)}; \theta\right) \tag{7}$$

where the expectation is taken with respect to the discrete measure derived from samples $\{\hat{X}_t^{(i)}\}_{i=1}^{n}$ from the generative model $\hat{P}_t$. The parameter estimation problem then requires optimizing both the generative model $\hat{P}_t$ and the networks $f$ and $\varphi$ representing the drift, while ensuring consistency between the two. Using knowledge of the PDE in (3), we regularize $\hat{P}_t$ such that it matches the flow relating to the drift. Additional details regarding the PDE and its relationship to the ML architecture are in the appendix.

## 4 Parameter Estimation

Having presented the relevant architectures, we now describe the procedures for estimating the parameters of the different architectures. We first describe the likelihood function for use in cases with regularly sampled data. We then describe a bridge estimator for cases of irregularly sampled data. Finally, we describe an estimator for the generative architecture based on both the likelihood function and the transition density. For this section, we assume that we observe multiple paths, i.e., $\left\{ \{X_{t_j}\}_{j=1...K}^{(i)} \right\}_{i=1...N}$. Full details of all algorithms are in the appendix.

### 4.1 Maximum likelihood estimation

We use an estimator based on the path-wise likelihood derived from Girsanov's theorem and an Euler-Maruyama discretization for the likelihood, considered in Sharrock et al. [2021]. The likelihood

function is given as

$$\mathcal{L}(\theta; t_1, t_K) := \exp\left(\frac{1}{\sigma^2}\int_{t_1}^{t_K} b\left(X_s, p_s, s; \theta\right) \mathrm{d}X_s - \frac{1}{2\sigma^2}\int_{t_1}^{t_K} b\left(X_s, p_s, s; \theta\right)^2 \mathrm{d}s\right). \quad (8)$$

Following discretization, with the approximations $\Delta X_{t_j} = X_{t_{j+1}} - X_{t_j}$ and $\Delta t_j = t_{j+1} - t_j$, the log-likelihood is approximated by

$$\log \mathcal{L}(\theta; t_1, t_K) \approx \sum_{j=1}^{K-1} b\left(X_{t_j}, p_{t_j}, t_j; \theta\right)\left(X_{t_{j+1}} - X_{t_j}\right) - \frac{1}{2}\sum_{j=1}^{K-1} b\left(X_{t_j}, p_{t_j}, t_j; \theta\right)^2\left(t_{j+1} - t_j\right).$$

If the time interval $\Delta t$ is large, then this likelihood loses accuracy, as is a property of the Euler-Maruyama discretization. Optimization is performed using standard gradient based optimizers with the drift $b$ represented as one of the presented architectures.

## 4.2 Estimation with Brownian bridges

Often data are not collected at uniform intervals in time, but rather, the time marginals may be collected at irregular intervals. In that case, we consider an interpolation approach to maximizing the likelihood following the results of Lavenant et al. [2021] and Cameron et al. [2021] in the Itô-SDE case. We can write the likelihood conditioned on the set of observations (dropping the particle index for ease of notation) as

$$\mathcal{L}_{BB}(\theta) = \mathbb{E}_{\mathbb{Q}}\left[\prod_{j=1\ldots K-1} \mathbb{1}\{Z_{t_{j+1}} - X_{t_{j+1}}\}\mathcal{L}(\theta; t_j, t_{j+1})\right]$$

where $\{Z_s : s \in [t_j, t_{j+1}]\}$ is a Brownian bridge from $X_{t_j}$ to $X_{t_{j+1}}$ and $\mathbb{Q}$ is the Wiener measure. Brownian bridges can easily be sampled and reused for computing the expectation, which reduces the variance of the estimator. By applying Jensen's inequality, we can write an evidence lower bound (ELBO) as

$$\log \mathcal{L}_{BB} \geq \mathbb{E}_{\mathbb{Q}}\left[\sum_{j=1\ldots K-1} \log \mathcal{L}(\theta; t_j, t_{j+1}) \ \middle| \ \left\{Z_{t_j} = X_{t_j}\right\}_{j=1}^{K}\right]. \quad (9)$$

The ELBO in this case aims to fit the observed marginal distributions exactly while penalizing deviations in regions without data that deviate from the Brownian bridge paths.

## 4.3 Estimation with explicit marginal law $\hat{P}_t$

Returning to the ML architecture described in Section 3.3, where we explicitly model the density $p_t$ with a generative network $\hat{P}_t$, our estimator should enforce the regularity of $p_t$ through its PDE in (3). Let the parameters of the drift be $\theta$ and the parameters of the generative model be $\phi$, then we solve the optimization problem

$$\max_{\theta, \phi} \quad \mathbb{E}\left[\mathcal{L}(\theta, \phi \mid \{X_{t_j}\}_{j=1\ldots K})\right] \quad s.t. \quad (10)$$

$$\int_{t_j}^{t_{j+1}} \left\|\hat{P}_s(x; \phi) - \mathbb{E}\left[\hat{P}_{t_{j+1}}\left(\hat{X}_{t_{j+1}}; \phi\right) \mid \hat{X}_s = x\right]\right\| \mathrm{d}s = 0 \quad (11)$$

for time intervals indexed by $j = 1\ldots K-1$, the state space $x \in \mathrm{supp}(X_t)$, and where the trajectories of $\hat{X}_t$ follow the dynamics of the ML architecture, specifically

$$\mathrm{d}\hat{X}_t = f(\hat{X}_t, t; \theta)\mathrm{d}t + \mathbb{E}_{y_t \sim \hat{P}_t(\cdot; \phi)}\left[\varphi\left(\hat{X}_t, y_t; \theta\right)\right]\mathrm{d}t + \sigma\mathrm{d}W_t. \quad (12)$$

The likelihood at the observed margins is first maximized in (10). In (11), the marginals at previous times are regularized using the correspondence between the PDE and its associated SDE via the nonlinear Kolomogorov backwards equation [Buckdahn et al., 2017], which describes $p_t$ as an expectation of trajectories at a terminal time, i.e. $p_t(x) = \mathbb{E}[p_T(X_T)|X_t = x]$ for $t < T$.

## 5 Modeling Properties

Having discussed the architectures and estimators, we now discuss specific properties of the modeling framework, which follow from the theoretical discussion presented in Section 2. We first discuss how the factorization into $\varphi$ and MF lends to an implicit regularization of the IM architecture. We then compare the gradient flows of Itô-SDEs and MV-SDEs.

### 5.1 Implicit regularization of the implicit measure architecture

Closely related to the IM architecture are neural Itô-SDEs, where we previously remarked can model MV-SDEs. On the other hand, the factorization of the IM architecture into $\varphi$ and MF leads to a type of implicit regularization when the parameters are estimated using gradient descent.

**Proposition 5.1** (Implicit Regularization). *Suppose $f$, $\varphi$ known and fixed. Further, assume that $\varphi$ is twice differentiable. Then, for each time step $t$, the minimizing finite width MF with weight matrix $W_0 \in \mathbb{R}^{n \times d}$ and ith row $W_0^{(i)}$ under gradient descent satisfies the following optimization problem*

$$\min_{W_0} \sum_{i=1...n} \sum_{j=1...d} \varphi(X_t, W_0^{(i)})_j \quad \text{s.t.} \quad \mathbb{E}\left[\frac{1}{2\Delta t} \|X_{t+\Delta t} - X_t - b(X_t, p_t, t)\|^2\right] = 0.$$

*Proof.* We follow the blueprint in Belabbas [2020] and give full details in the appendix. $\square$

Proposition 5.1 effectively says that the mean-field system approximated is the one that has the least influence from the other particles under perfectly matched marginals. In the case where $\varphi$ can be decomposed as a norm, this amounts to finding the drift parameterized by weight $W_0$ with smallest norm while still matching the marginals.

### 5.2 Gradient flows of the MV-SDE

To illustrate the difference between the MV-SDE and Itô-SDE particle flows, we invoke the analysis in Santambrogio [2017, Section 4.6] to describe the functionals that are minimized by each.

**Remark 5.2** (Functional Minimizer). *Consider two drifts $B = \nabla f(X)$ and $B_{MF} = B + \mathbb{E}[\nabla \varphi(X - y)]$. Consider a functional $F[p] = \int \log p \, dp + \int f(X) dp$ for some measure $p$ absolutely continuous with respect to the Lebesgue measure. Then, the gradient flow satisfying the linear Fokker-Planck equation with drift $B$ minimizes $F$. On the other hand, the nonlinear Fokker-Planck associated with drift $B_{MF}$ minimizes the functional $F_{MF}[p] = F[p] + \int \varphi(X - Y) dp(X) dp(Y)$.*

This has an important implication, for example, if we take $\varphi(\cdot) = 2\| \cdot \|\frac{dq}{dp} - \| \cdot \|^2 - \| \cdot \|^2 \left(\frac{dq}{dp}\right)^2$ then the functional is minimizing the squared energy distance between a target measure $q$ as well as the entropy. We use this example to motivate some of the experiments on probabilistic modeling.

## 6 Numerical Experiments

We discussed *Q1* on modeling and inferring distributional dependence. We now wish to answer *Q2* and quantify the effect of distributional dependence in machine learning tasks. To do this, we test the methods on synthetic and real data for time series estimation and sample generation. The main goal is to determine the difference between standard Neural Itô-SDE and the proposed Neural MV-SDEs under different modeling scenarios. In that sense, the baseline we consider is the Itô-SDE parameterized using an MLP. However, we also consider other deep learning based methods for comparison in a broader context. We abbreviate the different architectures as the Empirical Measure (EM) in Section 3.1, Implicit Measure (IM) in Section 3.2, and Marginal Law (ML) in Section 3.3. Full descriptions of the models, baselines, and datasets are given in the appendix.

**Synthetic data experiments** Motivated by the application of MV-SDEs in physical, biological, social, and financial settings, we benchmark the proposed methods on 4 canonical MV-SDEs: the Kuramoto model which describes synchronizing oscillators [Sonnenschein and Schimansky-Geier, 2013], the mean-field FitzHugh-Nagumo model which characterizes spikes in neuron activations

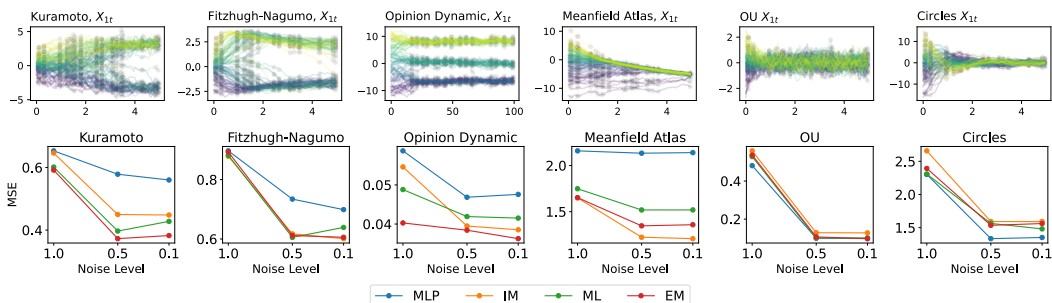

Figure 4: Top row: sample paths from the different synthetic datasets. Bottom row: mean squared error (MSE) of different architectures' performance on drift estimation, under the effect of different levels of observation noise. Reported value is an average of 10 runs.

[Mischler et al., 2016], the opinion dynamic model on the formation of opinion groups [Sharrock et al., 2021], and the mean-field atlas model for pricing equity markets [Jourdain and Reygner, 2015]. We additionally benchmark the proposed methods on two Itô-SDEs: an Ornstein–Uhlenbeck (OU) process and a circular motion equation to determine the performance on Itô-SDEs. Finally, to understand the performance on discontinuous paths, we benchmark the proposed methods on an OU process with jumps. We focus on recovering the drift from observations.

Since the true drifts of the synthetic data are known, we directly compare the estimated drifts to the true drifts. The performance on five different datasets with three different levels of added observational noise is presented in Figure 4. The proposed mean-field architectures outperform the standard MLP in modeling MV-SDEs; moreover, our experiments on OU and circular process suggest that incorporating explicit distributional depedence does not diminish the performance in estimating non-interacting Itô-SDEs. When modeling processes with

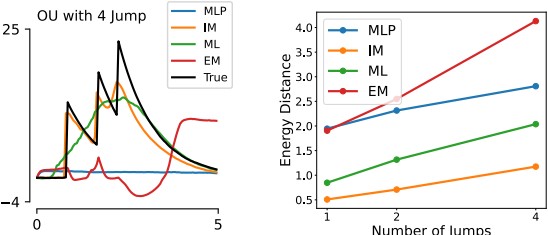

Figure 5: Left: Average paths of true and estimated OU process with 4 jumps. Right: Energy distance between true and generated paths.

jump discontinuities, Figure 5 highlights the flexibility of the proposed methods, IM, ML, to match such models. The EM likely does not perform as well due to the high variance of the empirical measure, leading to difficulties in learning. Additionally, the MLP does not have an explicit decomposition between the MV and Itô components, resulting in issues when estimating the feedback between the particles inducing jumps.

**Real data experiments** Extending from the synthetic examples, we consider two real examples: brain activity recorded by electroencephalograms (EEG), which is closely related to the Kuramoto model [Nguyen et al., 2020]; and chemically stimulated movement of organisms (chemotaxis), which can be modeled by the Keller-Segel model [Tomašević, 2021, Keller and Segel, 1971].

We evaluate the proposed architectures in these modeling tasks by comparing the goodness-of-fit of generated path samples to the observed path samples. We compute the Continuous Ranked Probability Score (CRPS) defined in Gneiting and Raftery [2007] (see appendix for details) for the 1-dimensional EEG data, and the normalized MSE (normalized with sample variance) for the 3-dimensional chemotaxis data with respect to the held out data. We also benchmark against the DeepAR probabilistic time series forecaster [Salinas et al., 2020] with RNN, GRU, LSTM, and Transformer (TR) backbones as another baseline model to compare the goodness-of-fit.

The performances of different architectures are presented in Table 1. For EEG, the proposed architectures generally perform better than the baselines in generating paths within the training time steps, and on par with the DeepAR architectures for forecasting (full results presented in appendix). For chemotaxis data, the MV-SDE based architectures all outperform the DeepAR baselines.

Table 1: Time series estimation on held out trajectories. NA/A stands for non-alcoholics/alcoholics. **Bolded** values and *italic* values are best and second best respectively.

|  | CRPS ↓ | | MSE ↓ | |
|---|---|---|---|---|
|  | NA-EEG | A-EEG | C.Cres | E.Coli |
| MLP (Itô) | 5.52 (1.40) | 4.33 (1.14) | 0.096 (0.002) | *0.080* (0.003) |
| IM | *5.23* (1.24) | 4.30 (1.21) | 0.094 (0.003) | **0.080** (0.001) |
| ML | **5.10** (1.22) | **4.05** (1.12) | **0.093** (0.002) | 0.084 (0.002) |
| EM | 5.35 (1.22) | *4.09* (1.11) | *0.093* (0.004) | 0.086 (0.004) |
| LSTM | 6.27 (2.02) | 5.68 (2.56) | 1.159 (0.234) | 0.585 (0.350) |
| RNN | 6.22 (2.07) | 4.64 (1.38) | 1.563 (1.070) | 0.773 (0.092) |
| GRU | 6.35 (2.01) | 6.18 (2.73) | 0.826 (0.289) | 0.568 (0.301) |
| TR | 5.95 (1.45) | 4.29 (1.36) | 1.503 (0.212) | 1.204 (0.212) |

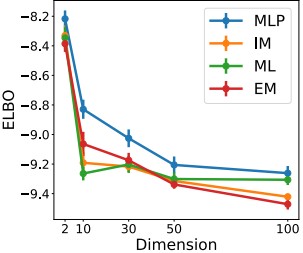

Figure 6: ELBO of generated paths from standard Gaussian to eight Gaussian mixture (in increasing dimension) evaluated against OT mapping.

**Generative modeling experiments** We focus on applying the bridge estimator discussed in Section 4.2 to map between a Gaussian and a target distribution. We are interested in two aspects: 1) the properties of the learned mapping, and 2) the generated trajectories. We first study the properties of the learned mapping using a synthetic eight Gaussian mixture with increasing dimensionality. We compare the performance of different architectures through the ELBO of the sample paths generated by the optimal transport (OT) mapping between the initial distribution and held out target samples. We next evaluate the generated trajectories through the energy distance (see appendix for details) between generated and held-out data for 5 real data density estimation experiments. In addition, we compare to common density estimators of variational autoencoder (VAE) [Kingma and Welling, 2013], Wasserstein generative adversarial network (W-GAN) [Gulrajani et al., 2017], masked autoregresive flow (MAF) [Papamakarios et al., 2017] and score-based generative modeling through SDEs, which corresponds to a constrained form of the MLP [Song et al., 2020]. The MV-SDE architectures not only outperform the Itô architecture for all dimensions in the eight Gaussian experiment, as shown in Figure 6, but also for the 5 real data density estimation experiments, as shown in Table 2, while outperforming common baselines. All sampling is performed using standard Euler-Maruyama, with full details of the sampling and inference algorithms in the appendix. This again suggests the MV-SDE provides a more amenable probability flow for modeling compared with the Itô case.

Table 2: Density estimation: Energy distance between observed samples and generated samples of different methods. **Bolded** values and *italic* values are best and second best correspondingly.

|  | POWER | MINIBOONE | HEPMASS | GAS | CORTEX |
|---|---|---|---|---|---|
| MLP (Itô) | 0.342 (0.096) | 0.674 (0.048) | 0.537 (0.052) | 0.405 (0.08) | 0.742 (0.062) |
| IM | 0.292 (0.078) | **0.395** (0.045) | 0.405 (0.025) | **0.287** (0.082) | **0.53** (0.026) |
| ML | **0.282** (0.083) | *0.443* (0.034) | *0.366* (0.03) | 0.305 (0.063) | 0.568 (0.03) |
| EM | 0.328 (0.116) | 0.455 (0.036) | 0.429 (0.046) | *0.298* (0.036) | 0.577 (0.037) |
| VAE | 1.19 (0.024) | 2.117 (0.148) | 1.763 (0.031) | 1.516 (0.023) | 2.412 (0.197) |
| W-GAN | 1.248 (0.017) | 2.079 (0.003) | 1.819 (0.013) | 1.3 (0.016) | 2.19 (0.011) |
| MAF | *0.288* (0.041) | 0.467 (0.009) | **0.308** (0.017) | 0.519 (0.033) | *0.532* (0.026) |
| Score-Based | 0.302 (0.049) | 0.499 (0.019) | 0.324 (0.028) | 0.562 (0.043) | 0.582 (0.020) |

## 7  Discussion

In this paper we discuss an alternative viewpoint of the standard Itô-SDE parameterization. In particular, we focus on MV-SDEs and discuss how neural networks can represent a process that depends on the distribution, and we describe ways of making this dependence more explicit. We demonstrated the efficacy of the proposed architectures on a number of synthetic and real benchmarks. The results suggest that the proposed architectures provide an improvement over baselines in certain generative modeling and time series applications.

**Limitations** We only studied the implicit regularization of the IM architecture under gradient descent, but the extension of the analysis to the other proposed architectures is important to understand the corresponding regularization. Additionally, computing expectations incurs additional computational cost. Improving the computational accuracy using a multilevel scheme as proposed in Szpruch et al. [2019] could improve the performance of the methods.

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
