# OpenReview forum: "Neural McKean-Vlasov Processes: Inferring Distributional Dependence"
_NeurIPS.cc/2023/Conference — Submitted to NeurIPS 2023_

### Official Review · Reviewer_bDXo · 2023-06-25

**Soundness:** 4 excellent
**Presentation:** 4 excellent
**Contribution:** 3 good
**Rating:** 8
**Confidence:** 3

**Summary:**

The submission considers McKean-Vlasov Stochastic Differential Equation models, which are a generalization of the more-familiar Ito processes. The difference is that the former additionally feature the evolution of the law of the process $X_t$ (denoted $p_t$) as well as $X_t$ itself. Such processes are the limit as the number of particles in a system approach infinity, and their finite-particle approximation (where $p_t$ becomes some empirical distribution) give rise to a model that exhibits temporal as well as between-particle interactions.

A key aspect of the work is to posit a model whereby $p_t$ enters only through a term $\mathbb{E}_{y_t \sim p_t}[\varphi(X_t, y_t)]dt$ for some *interaction function* $\varphi$. The advantage of such a formulation is that the transition density of the process then reduces to the solution of a PDE.  The work discusses the properties of MV-SDEs of the aforementioned form that make them a desirable class of models (e.g., non-local interactions between paths, and the ability to incorporate jumps).

Three neural architectures are proposed, differing in particular for their approach as it relates to the modelling of $p_t$:
* Implicit Measure: which recasts the empirical measure as a single layer of a neural network (though I did not completely understand how this happens). The implicit measure architecture has the advantages of being able to cope with the missing data setting.
* Empirical Measure: where $\varphi(\cdot, \cdot)$ takes the form of a trainable neural network.
* Marginal Law: which involves learning a generative model.

Methods for parameter estimation are presented (Section 4), including methods for the missing data setting based on Brownian bridges. A result is proven (Proposition 5.1) regarding implicit regularization properties. A numerical study is conducted on a number of simulated data examples, real data (for which forecasting is also explored), as well as a generative modelling task.



**Strengths:**

* The ideas contained in the submission are highly non-trivial, yet explored in impressive depth from a number of different angles in a sophisticated manner.
* The study of such techniques is impressively comprehensive in terms of both theory and different approaches (giving a number of methodological approaches, as well as some theory). The submission appears to represent the outcome of a significant body of work and investigation.
* The submission is very well written and presented. The numerical experiments appear well-executed.

Despite some familiarity with concepts in the submission, I am not an expert on SDE modelling, so can not particularly comment on the novelty in that regard. However, the ideas in the submission appear like ones that are very natural to explore, have been explored well, and are worthy of publication in my opinion.


**Weaknesses:**

* As much as I enjoyed the abstract presentation, it would be beneficial to have additional clarity as to settings where MV-SDE modelling may be of interest, so to contextualise the work, or where different approaches would be preferred. It is mentioned that few works have considered such models in machine learning tasks, and it would be beneficial to have a small discussion what the contribution could potentially be there (at a high level, that is).

* Related to the above, a clearer motivation of the task at hand would be beneficial. The paper takes an SDE-first viewpoint, but isn't the overall goal to fit some sort of particle approximation to the SDE? Some additional discussion and background would be beneficial.

* Some applications are mentioned at present, but it would be nice to have a small discussion something along the lines of "MV-SDEs are most useful when the goal of interest is to model... ".

* The derivation of the implicit measure architecture was something that I could not properly parse as currently written.



**Questions:**

* The simplified case of fixing the diffusion coefficient is considered, and only looking at learning the drift. How strong is this restriction in terms of modelling potential? Does it exclude certain behaviour?
* Similar to the above, but for the factorization imposed on $b(X_t, p_t, t)$ (though I do understand the impetus for such is the PDE representation, I am curious about the other aspects).
* p4, l130: Additional background should be given regarding what "mean-field drift" is, and mean-field approximations of MV-SDEs in general. Those without a strong background in MV-SDE modelling will likely not understand the paper otherwise. Also, should "in (2)" not be "of the form in (2)", as (2) is the limiting case?
* p4, l153: It is not clear what the intended meaning of this equation or the sentence preceding it is. Is the intention to say that one is using an MLP with stochastic neurons?  This section would benefit for additional details and clarity.
* p4, l151: Why does having few samples make it difficult to obtain an empirical measure? I would have thought this would simply made the computation easier.
* Regarding the title: "inferring distributional dependence" is a little vague, and perhaps could be replaced by something else that does the paper more justice and gives those without knowledge of what a MV-SDE is more of a clue what the paper is about (this is a very minor point, and just a suggestion).

Minor things:
 * p5, l180: should $(X,p)$ be $(X_t, p_t)$?

**Limitations:**

A little additional discussion would be beneficial (see other parts of this review for specifics).

---

> ### Author Rebuttal · Authors · 2023-08-09
>
> We thank the reviewer for the detailed and invaluable comments.
> We also appreciate the positive remarks.
> We address the individual concerns below.
>
> 1. (Where MV-SDEs may be of interest)
>
> The reviewer makes a good point, we will include additional discussion on the applications of MV-SDEs and their comparison to Ito-SDEs.
>
> We will add the following note:
>
> Ito-SDEs do not include interactions with other sample paths and is appropriate when modeling dynamics that are known to be independent,
> such as the movement of molecules that do not interact.
> On the other hand, it is often natural that the dynamics of particles influence each other.
> There are many examples where this interaction is important to model.
> It is well illustrated in a double well system that has two stable states. In the It\^o case with no particle interaction, the probability of a particle switching states is exponentially small.
> In the McKean-Vlasov case, particles can switch potentials through the influence of their mean-field interactions as shown in [1].
> As another example, modeling the beliefs of different agents who can influence the opinions of other agents, leading to polarization.
> Additionally, MV-SDEs have appeared within the context of analyzing the behavior of neural networks, specifically the transformer architecture [2].
> MV-SDEs also appear in inferring the trajectories of single cell RNA data [3].
> The interactions between cells are important to model to maintain dynamics that are similar to the data.
>
> [1] Garnier et al. "Large deviations for a mean field model of systemic risk." SIAM Journal on Financial Mathematics 2013.
>
> [2] Sander et al. "Sinkformers: Transformers with doubly stochastic attention." AISTATS 2022.
>
> [3] Chizat et al. "Trajectory inference via mean-field langevin in path space." NeurIPS 2022.
>
> 2. (Clearer task motivation and applications)
>
> The reviewer makes a good point on the motivation.
> While we started motivating with the two questions within the introduction, we should include additional context on how these questions apply.
> We will rewrite the introduction to include the following:
>
> "
> In many scientific disciplines, the interaction of different agents is important to study and inferring these interaction properties from data remains a challenging task.
> Curiously, similar ideas have also found their way in probabilistic modeling in machine learning settings but have not been very well studied through the lens of particle interactions.
> For example, when inferring single cell RNA trajectories, McKean-Vlasov processes have been used to find the correct particle distributions [3].
> Our goal is then to identify appropriate techniques for applying McKean-Vlasov processes in both (scientific and machine learning) disciplines and study their influences on the respective problems.
> "
>
> [3] Chizat et al. "Trajectory inference via mean-field langevin in path space." NeurIPS 2022.
>
> 3. (Derivation of implicit measure architecture)
>
> We apologize for the confusion here.
> Please refer to the mean-field layer rewrite in the response to all the reviewers.
>
> 4. (Fixed diffusion coefficient)
>
> A fixed diffusion coefficient generally implies that the marginal density has roughly exponential decay.
> The model and algorithms can work with an estimated or known constant diffusion coefficient (it only requires a small change in the ELBO to estimate this parameter).
> Since we would like to focus on estimating the unknown drift which is related to process trend, we assume a known constant diffusion coefficient.
> We thank the reviewer for pointing this out and we will include a discussion on estimating the diffusion coefficient in the appendix.
>
> 5. (Factorization of the drift)
>
> We thank the reviewer for pointing these out and added a discussion on the factorization in the response to all reviewers.
>
>
> 6. (MLP architecture exposition)
>
> We apologize for the confusion here.
> The main point regarding this paragraph is to show that we can write $\mathbb{E}_{y \sim p_t} [\varphi(x - y)]$ as an MLP where the width tends to infinity and using an activation function given by $\varphi$.
> To see this, we need the weight to correspond to a delta function at the point $x$ and the bias to correspond to many samples from $p_t$.
> This motivates why it is natural to consider a MLP for the task of estimating MV-SDEs and motivates how to build upon this idea to develop more general architectures to support MV-SDEs.
>
> The architectures section begins with these thoughts on the MLP architecture and then continues by considering different modifications of the base MLP structure and their influences on the performance in some of the tasks of interest.
>
> 7. (Few samples in the empirical measure architecture)
>
> The reviewer is correct that having only a few samples would make the computation easier (since the summation over a smaller number of points in the expectation would be easier to compute).
> However, when considering too few samples, the estimation of the expectation may be incorrect.
> Consider the case of a double well potential where at some time marginals we only observe samples from one of the potentials.
> The empirical measure in this case would be biased to one particular mode of the distribution for that time marginal.
>
>
> 8. (Title changes)
>
> This is a good point, we mainly want to emphasize that the architectures that provide the decomposition in terms of the MV-SDE includes distributional dependence.
> We can possibly change it to "Distributional dependence in diffusion models", but we are open to other suggestions that the reviewer may have.
>
> 9. $(X, p) \to (X_t, p_t)$
> The reviewer is correct, and we would like to thank the reviewer for pointing this out, we will correct it in the final manuscript.

---

> > ### Comment · Reviewer_bDXo · 2023-08-11
> >
> > Many thanks to the authors for their thorough response. I confirm that I have read it as well as the other reviews and their responses. I commend the authors not only on the details of their response, but in providing concrete examples of new explanations.  I retain my score of "Strong Accept" as a result, though in light of the clarifications now increase my confidence score from two to three.
> >
> > Regarding two minor points:
> > * (MLP architecture exposition) Thanks for the clarification. It may be beneficial to include it in the final paper, but this is at the authors discretion.
> > * (Title) I agree and understand the motivation of the title in the original submission, and confess I don't have any particularly better suggestions. This was a very minor point, and entirely up to the authors. I like the alternative being considered, but it may be best not to say "diffusion models" as, while entirely correct, the term tends to be more associated with a particular class of deep generative models these days. Perhaps the original title is indeed optimal after all!

---

> > > ### Author Response · Authors · 2023-08-11
> > >
> > > Many thanks to the reviewer for the kind and fast reply. We will include the clarification on the MLP architecture in the final paper. We agree that diffusion model tends to be more associated with a particular class of deep generative models and it may be good to retain the focus on MV-SDEs and inferring distributional dependence. Nevertheless, we would like to thank the reviewer for helping to bring up the suggestion of a title change. Once again, thank you very much for all your comments and feedback!

---

### Official Review · Reviewer_Ev1V · 2023-07-10

**Soundness:** 1 poor
**Presentation:** 1 poor
**Contribution:** 1 poor
**Rating:** 2
**Confidence:** 4

**Summary:**

This paper considers the problem of parameter estimation from data when the underlying dynamical system is modeled by the MV-SDE. To represent the target MV-SDE, the authors propose two strategies: (i) expressing a layer in a neural network as an expectation with respect to a density and (ii) using generative models to capture distributions that generate observations at different time stamp. With these strategies, the authors then propose to conduct parameter estimation via 1. maximum likelihood estimation, 2. using Brownian bridge for estimation, 3. explicit marginal law estimation.


**Strengths:**

Please see the discussion below.

**Weaknesses:**

1. Poor presentation. Overall this paper is obscure and hard to follow. Here are some detailed examples.
* When stating the underlying MV-SDE model in Eq.(2), it is not clear which terms are known and which terms are to be learned. Specifically, do we know $f$ and $\phi$? Since we are interested in the problem of parameter estimation, both terms should be learned from the observations.
* It is now clear how the proposed implicit measure architecture can be carried out: In Eq.(6), it is not clear what $\mathbb{P}_t$ and $\mathbb{P}_0$ are and why we can estimate the Radon–Nikodym derivative $d \mathbb{P}_t/d\mathbb{P}_0$. Moreover, the authors are motivating the implicit measure architecture as an alternative to the standard empirical measure approach since it can handle the situation when only samples from irregular time stamps are available. However, it is not clear why this is the case.
* In section 4.2, where the authors propose to estimate parameters using the Brownian bridge, it is not clearly how the Brownian bridge comes into play.
* The maximum likelihood estimation in section 4.1 is proposed in previous work [Sharrock et al. 2021], but is presented as a strategy proposed by this work, which is misleading.

2. Lack concrete contributions. Partly due to the poor presentation of this work, this paper presents no clear contributions. For example, the proposed implicit measure architecture and the marginal law architecture are just two ways to rephrase the standard empirical measure.  The results presented in section 5 are well-known results in the literature, e.g. the Wasserstein gradient flow structure of the MV-SDE.

3. A contradiction between the goal of this project and its fundamental assumption. While the authors motivate the research of MV-SDE to model jump (discontinuous behavior) in time sequence data, they also assume that the drifting term of the MV-SDE is sufficiently regular. This is a clear contradiction. For MV-SDE with regular drifting term and interaction term, it can be proved that the characteristic flow is also regular.



**Questions:**

Please see the comments above.

**Limitations:**

Please see the comments above.

---

> ### Author Rebuttal · Authors · 2023-08-09
>
> We regret that the reviewer felt so negatively towards the work and could not find a single strength within the entire manuscript.
>
> 1. (Poor presentation)
>
> We regret the reviewer found the presentation poor.
> We will address the reviewer's points.
>
> - Both $f$ and $\varphi$ are learned.
> We would like to bring to attention in lines 89, 90, we stated that we ''focus on estimating the drift, $b$, from data'', then proceeded to decompose $b$ into $f$ and $\varphi$ in equation (2). In lines 136, 137, we stated that we `` denote a function $f$ parameterized by parameters $\theta$ as $f(\cdot;\theta)$ then proceeded to note in lines 142-144 that $f(\cdot;\theta)$ and $\varphi(\cdot;\theta)$ are represented with neural networks.
>
> - $P_t$ is the marginal distribution at time $t$, $P_0$ is the base measure at an arbitrary time (call it `$0$').
> Both are absolutely continuous with respect to the Lebesgue measure and the Radon-Nikodym derivative exists.
> Unlike the EM architecture, the IM architecture estimates the time marginal law implicitly through a change of measure with the Radon-Nikodym derivative, which means we do not need a different set of samples at each time point.
> For further details, please also refer to the mean-field layer exposition in the response to all reviewers.
>
> - The Brownian bridge is used to sample paths between observed time margins for data collected at irregular time intervals.
> We would like to bring to attention that the application of Brownian bridges to irregular time intervals as an interpolator in maximum likelihood estimation was stated in lines 208-210.
>
> - The reviewer's comment that we present Sharrock et al's work as ours is *not true*.
> First, we clearly cited the work in line 200.
> Second, the proposed approach by Sharrock et al involves computing the stochastic exponential, which is a standard approach within most SDE parameter estimation techniques.
> Third, our contribution is within the context of the relevant problems within the NeurIPS community, which Sharrock et al did not consider.
> Out of the many works that describe SDE parameter estimation, we cited Sharrock et al's work due to its focus on MV-SDEs.
>
> 2. (Lack of concrete contributions)
>
> We regret the reviewer found our work lacking in concrete contributions.
> The contributions of this work were clearly delineated in the introduction. The reviewer even described our contributions within their summary at the beginning of the review.
> We restate that our main contributions are the architectures and the analysis while providing an estimation framework within the context of the architectures.
> The analysis we provide linking the approximation capabilities have not been considered in the literature.
> The analysis implies that modeling MV-SDEs using neural networks is appropriate and then considers specific architectures and their implications.
>
> The reviewer's comment that ``the proposed implicit measure architecture and the marginal law architecture are just two ways to rephrase the standard empirical measure'' is incorrect. The standard empirical measure architecture relies on observations to compute the empirical expectation, the proposed architectures learn the measure from observations.
>
> The reviewer's comment that ``the results presented in section 5 are well-known'' is incorrect. The implicit regularization analysis in section 5.1 is new and specific to this work while the application to the energy distance in 5.2 has not been done before (as far as we are aware).
>
> To say that these are not worthy contributions would negate a large number of contributions within the machine learning literature, since many results pull from existing topics and consider them within a new context.
> For example, the mathematics behind diffusion models is well known yet its introduction within the machine learning community has led to many advances in probabilistic modeling.
> As such, we strongly disagree with the reviewer on this point.
>
>
> 3. (Contradiction between stated goals and assumptions)
>
> We assumed that the drift is sufficiently regular in section 2.1, where we established the background. As is often done for the analysis, we impose restrictions on the processes to make the exposition and theoretical developments clear.
> We then described discontinuous sample paths in section 2.3 to illustrate and motivate an interesting property of MV-SDEs which specifically is due to the interactions between particles. We will note in the discussion of the motivations, experiments (and specifically the mean-field atlas model and OU jump process), and limitations that we relax these assumptions and consider the performance of the proposed architectures on a wide range of scenarios that may not satisfy the assumptions.
> This is a strategy in many machine learning papers, since in many scenarios it is often impossible to determine whether assumptions are satisfied.
> This does not imply a contradiction, since often algorithms are applied in scenarios that have assumptions that are not verifiable or to demonstrate how they work beyond the scope of the original assumptions.
> This is hardly a reason to reject a paper, but rather it showcases the versatility of the methods beyond the original theoretical constraints.
>
> Separately, though not path discontinuities, we would like to bring to attention an interesting and related case of phase transitions with only interaction through weak attraction. Simulation parameters and proofs are given in [1].
> We would also like to bring to attention that for the case of positive feedback, under relaxed assumptions on the drift, a simple proof is given in [2] Theorem 1.1 that the path is discontinuous.
>
> [1] Garnier et al. "Large deviations for a mean field model of systemic risk." SIAM Journal on Financial Mathematics 2013.
>
> [2] Hambly et al. "A McKean-Vlasov equation with positive feedback and blow-ups." Annals of Applied Probability 2019.

---

> > ### Comment · Reviewer_Ev1V · 2023-08-16
> > **Response to the authors' rebuttal**
> >
> > First of all, I thank the authors for the detailed response.
> >
> > Regarding to the presentation of the IM architecture, I think now I understand the meaning of proposed approach. Please let me know if the following restatement of the approach is correct:
> >
> > The purpose of this this "layer" is to approximate the mean-field interaction $\phi \ast p_t$. Since we have $\phi \ast p_t(x) = \int \phi(x, y) \frac{p_t}{\mu}(y) \mu(y) d y$, the author proposes to approximate this quantity by $\int \phi(x, y; \theta) h(t, y; \xi) \mu d y$. Here, $h(t, y; \xi)$ is some neural network parameterized by $\xi$ and is used to approximate the time-varying importance weight $\frac{p_t}{\mu}$. Further, the authors then decide to merge $\phi(x, y; \theta)$ and $h(t, y; \xi)$ and define this quantity as $\phi(\cdot, \cdot, t; \theta)$. With the abuse of the notations $\phi$ and $\theta$ ($\phi$ and $\theta$ defined in line 165 are of different meaning from the ones mentioned in Eq.4), I could not parse the sentence in line 165-166.
> >
> > Should my understanding be correct, I guess for the easy of the audience, the authors should very clearly state what is the quantity to be approximated using neural network and what are the parameters to be learned.
> >
> >
> > In terms of the novelty of the proposed approach, I do not find the IM architecture and the ML architecture new as they are simply the importance weight and push-forward model commonly used in the ML community to represent distributions.
> >
> > >  the energy distance in 5.2 has not been done before (as far as we are aware).
> >
> > Please check the diffusion-advection-interaction equation in Eq.(4.14) of [Santambrogio 2017]. This is a paper cited by the authors and I am surprised that the authors did not know this result.
> >
> > > We would also like to bring to attention that for the case of positive feedback, under relaxed assumptions on the drift, a simple proof is given in [2] Theorem 1.1 that the path is discontinuous.
> >
> > Interesting. I apologize for not noticing this work as you have cited it in line 126. However, I find it hard to formulate the SDE considered therein as an instance of the MV-SDE in Eq.(2) considered in your work. Could you please elaborate on this?

---

> > > ### Author Response · Authors · 2023-08-17
> > >
> > > We thank the reviewer for the response, below we address the three main comments.
> > >
> > > 1. The reviewer's understanding of the IM architecture is now correct.
> > >
> > > We note that a large part of the field of machine learning uses similar notions of importance sampling and push forward measures in many research directions, so the reviewer's questioning of the novelty of our paper because we also applied these techniques is surprising.
> > >
> > > Additionally, we note a few of the other contributions that are new:
> > > First, we study the effect of distributional dependence on generative modeling and time series.
> > > This is a new direction and, as highlighted in the text, has high relevance due to research interest in diffusion models and the relationship to attention.
> > > Second, the proposed architectures provide a practical way of applying distributional dependence to the problems of interest and empirically demonstrate improvements over existing methods.
> > > Third, the analysis of the IM architecture provides intuitive properties on the regularization induced by the architecture.
> > > This allows a user to gain an idea of how the process is learned using this particular architecture.
> > > In that sense, we believe these are valuable contributions to the machine learning community, with particular emphasis to those working on generative modeling using diffusions.
> > >
> > >
> > > 2. We regret that the reviewer may be mixing the energy distance with energy functions. These are not the same. The equation 4.14 in Santambrogio 2017 describes an energy function related to the granular media equation.
> > > However, we must emphasize that we are describing a specific functional that minimizes the *energy distance* between two probability distributions, *not an arbitrary energy function*.
> > >
> > > We are not aware of an existing reference where the energy distance has been studied or derived within the context of gradient flows.
> > > We then use this to motivate the experiments where we study how the different architectures minimize the energy distance with respect to a target distribution.
> > > We would be happy to cite a reference if the reviewer knows of one regarding the energy distance.
> > >
> > > 3. We can represent this with drift $-\alpha  \frac{\partial}{\partial t}\mathbb{E}[\mathbf{1}_{y_t \leq 0} ]$, $\alpha\in\mathbb{R}^+$.

---

> > > > ### Comment · Reviewer_Ev1V · 2023-08-17
> > > > **Response to the authors' rebuttal**
> > > >
> > > > I thank the authors for the quick response.
> > > >
> > > > > We regret that the reviewer may be mixing the energy distance with energy functions. These are not the same. The equation 4.14 in Santambrogio 2017 describes an energy function related to the granular media equation.
> > > >
> > > > I am sorry, but I could not identify the difference between the `energy distance` and the `energy functions` in the context of your paper. Could you elaborate a bit on this?
> > > >
> > > > > However, we must emphasize that we are describing a specific functional that minimizes the energy distance between two probability distributions, not an arbitrary energy function.
> > > >
> > > > My point here is that the conclusion in Remark 5.2 is a standard results in the literature of Wasserstein gradient flow, which is supposed to be general. I am not sure why deriving the same result for a specific instance adds additional novelty to the generic result.
> > > >
> > > > > We can represent this with the drift ...
> > > >
> > > > Sorry, could you be more specific? In particular, what are the choices of $f$ and $\phi$ to take in order to recover the dynamics of [2]? Thanks!

---

> > > > > ### Author Response · Authors · 2023-08-17
> > > > >
> > > > > The energy function is a general term that is minimized according to the gradient flow.
> > > > > The energy distance is a statistical quantity that measures the distance between two probability distributions (please see [1,2]).
> > > > > We present the remark such that we can show that the flow minimizes the energy distance, which we derive for this case.
> > > > >
> > > > > Note that the remark is not a lemma or theorem and not claimed to be something novel, we are using this remark in deriving the energy distance which is the component we believe has not been derived elsewhere.
> > > > >
> > > > > Finally, take $f(x_t) = 0$ and $\varphi(x_t,y_t) =  \mathrm{ReLU}(y_t) / y_t - \mathrm{ReLU}(y_{t-\Delta t}) / y_{t-\Delta t}$ following discretization in time.
> > > > >
> > > > > [1] Szekely, Rizzo (2013) Energy Statistics: A Class of Statistics Based on Distances
> > > > >
> > > > > [2] https://en.wikipedia.org/wiki/Energy_distance

---

> > > > > > ### Comment · Reviewer_Ev1V · 2023-08-17
> > > > > > **Response to the authors' rebuttal**
> > > > > >
> > > > > > > We present the remark such that we can show that the flow minimizes the energy distance, which we derive for this case.
> > > > > >
> > > > > > Sorry, but I still do not understanding what is derived in this work in the regard of `energy distance`. Could you provide a concrete statement of your result? Line 255 is too brief to parse. What is $q$ therein?
> > > > > >
> > > > > > > Finally, take ...
> > > > > >
> > > > > > Thanks. While it might seems straightforward to the authors, I am not clear how is choice of $\phi$ gives the dynamics in [2]. Could you provide a derivation for this result?

---

> > > > > > > ### Author Response · Authors · 2023-08-18
> > > > > > >
> > > > > > > We want to show there exists a $\varphi$ such that the gradient flow under this $\varphi$ leads to the minimization of the energy distance between two densities $p,q$.
> > > > > > > The energy distance is defined as:
> > > > > > > $$
> > > > > > > E(p, q) = 2\int \int \| x - y \| dp(x) dq(y) - \int \int \|x - x'\| dp(x) dp(x') - \int \int \|y - y'\| dq(y) dq(y')
> > > > > > > $$
> > > > > > > between two densities $p, q$.
> > > > > > > Assume that $q \ll p$ so that the Radon-Nikodym derivative exists.
> > > > > > > By linearity of expectation, we can rewrite as
> > > > > > > $$
> > > > > > > E(p, q) = \int \int \left ( 2\| x - y \|  \frac{dq}{dp} - \|x - x'\|  -  \|y - y'\| \left(\frac{dq}{dp}\right)^2 \right)dp(x) dp(x').
> > > > > > > $$
> > > > > > > This is now in a form that is usable in the original gradient flow formulation.
> > > > > > > We then choose the function in the parentheses as $ \varphi = 2\| x - y \|  \frac{dq}{dp} - \|x - x'\|  -  \|y - y'\| \left(\frac{dq}{dp}\right)^2$ to minimize the energy distance.
> > > > > > >
> > > > > > > We encourage the reviewer to check equation 1.2 in the Hambly reference to see a related expansion of the expectation in terms of an indicator function.
> > > > > > > $\tau$ is the hitting time at 0 and $P(\tau < t)$ is the CDF of the hitting time evaluated at the current time, which relates to how many particles are less than or equal to zero at the current time.
> > > > > > > This is equivalent to the integral of the PDF of the particles less than zero, which is given by the expectation of the indicator function on particles less than zero.
> > > > > > > We can write the indicator function of a value as the ReLU function of the value divided by the value itself to obtain the statement in the previous response.

---

> > > > > > > > ### Comment · Reviewer_Ev1V · 2023-08-18
> > > > > > > > **Response to the authors' rebuttal**
> > > > > > > >
> > > > > > > > > This is now in a form that is usable in the original gradient flow formulation.
> > > > > > > >
> > > > > > > > OK. Could you explain how a gradient flow that minimizes the energy distance relates to this paper? For example, is your algorithm related to this gradient flow? If so, please point out this relation and clarify it in the revision. I only found the key word `energy distance` used as a quality metric for the output solution in the experiment section, but could not understand what is the role of the gradient flow.
> > > > > > > >
> > > > > > > > >  We encourage the reviewer to check equation 1.2 in the Hambly reference to see a related expansion of the expectation in terms of an indicator function.
> > > > > > > >
> > > > > > > > Sorry, I cannot really understand the statement in your previous response. Here, the interaction kernel $\phi(x, y)$ should be some time-independent function of $x$ and $y$. Moreover, for the purpose of representability by a neural network, $\phi$ should be at least continuous w.r.t. both $x$ and $y$. However, in your statement, this quantity is independent of $x$ and seems to be time dependent (I am not sure what is the meaning of $\Delta t$ when you are defining the function $\phi$).
> > > > > > > >
> > > > > > > > To address my major concern (the contradiction between the goal of this project and its fundamental assumption), the authors need to provide an example MVE such that its interaction kernel is sufficiently regular, e.g. continuous, but the dynamics is discontinuous after a finite evolving time. The best practice would be to clearly state the example and explain why this happens.

---

> > > > > > > > > ### Author Response · Authors · 2023-08-19
> > > > > > > > >
> > > > > > > > > In the generative modeling experiments, we sample from a target distribution that is given by a probability flow governed by a MV-SDE.
> > > > > > > > > Using the energy distance statement, one can see that a particular type of the MV-SDE can be used to minimize the energy distance to a target distribution (that is, it can be learned such that the flow minimizes the energy distance for all time steps).
> > > > > > > > >
> > > > > > > > > We provide another example that may be easier to see from the reference [1] which describes another MV-SDE that exhibits jumps.
> > > > > > > > > The drift in this case would be given by $E[1_{Y_t > 1}]$ such that the number of hits of a boundary are counted.
> > > > > > > > > The kernel in this case is $\varphi(x,y) = 1_{y > 1} + 0 x$.
> > > > > > > > > $\mathrm{ReLU}(y - 1)/(y-1)$ can again be used to represent the indicator. The idea is that as particles enter the region of interest, a feedback mechanism comes into play that results in a jump discontinuity.
> > > > > > > > >
> > > > > > > > > Note again this is not the overall goal of the project but a motivation on the properties of MV-SDEs and why they are interesting, as we already described the main goals and contributions of the paper outside of this example. Please refer to our original response to your review in point 3.
> > > > > > > > > The subscript $t$ in $y_t$ refers to the expectation at the $t$ time marginal.
> > > > > > > > >
> > > > > > > > > [1] Delarue, François, et al. "Global solvability of a networked integrate-and-fire model of McKean-Vlasov type." Annals of Applied Probability

---

### Official Review · Reviewer_DRq5 · 2023-07-10

**Soundness:** 2 fair
**Presentation:** 3 good
**Contribution:** 3 good
**Rating:** 6
**Confidence:** 3

**Summary:**

The authors proposed two new methods of modelling McKean--Vlasov SDEs using a neural network, and studied its empirical performance.

Since I'm not an expert this exact topic, I would like to ask the authors some questions first. I would be happy to raise my score further once I understand the paper better.

**Strengths:**

The authors proposed methods that do not model a finite population of the particles, which is a very interesting alternative.

**Weaknesses:**

Several parts of the paper seem unclear to me at the moment, and I will ask specific questions next.

**Questions:**

Again, I would be happy to raise my score once my questions are adequately addressed.

1. Let me start with a basic question: under what type of conditions can we factor the drift $b$ into $f$ and $\varphi$ components as in equation 2? This seems to be an important simplification and I would like the authors to motivate this further. Perhaps the authors can provide some important examples in applications where this simplification is available.

2. The authors seem to also occasionally write $\varphi(x,y) = \varphi(x-y)$ fairly interchangeably, e.g. in equation 3. Can the authors clarify if this is intentional in any of the circumstances? For example, it seems like in the MLP representation case of Appendix A.1, it is necessary to use $\sigma = \varphi(x-y)$.

3. On a related note, when is $\varphi$ known a priori? It seems like if this is known, then we wouldn't need a neural network to estimate $\varphi$ in the empirical measure approach, we would just directly simulate the particle system?

4. The authors wrote that MLPs can model McKean--Vlasov dynamics, but in the derivations it seems like it would require the weights to be identity matrix, which fundamentally restricts the complexity of the bias $b$ since it can only be $d$-dimensional. So doesn't this mean that $\nu$ can at most be an average of $d$ Dirac-delta's, and not necessarily capable of representing a general distribution $p_t$?

I found the definition and discussion about the mean field layer quite confusing. I have several questions specifically dedicated to this.

5. What is the distribution $\mathbb{P}_t$, and how does the Radon--Nikodym derivative $\frac{d \mathbb{P}_t}{d \mathbb{P}_0}$ show up in equation 6?

6. When written along with $\varphi( X_t, W_0^{ (i) } )$, is the Radon--Nikodym derivative a function of $X_t$ or $W_0^{(i)}$? As in which measure is being changed here?

7. Can we interpret each of the $W_0^{(i)}$ as a hypothetical particle?

8. Most importantly, the authors suggested that the Radon--Nikodym derivative can be learned, and that as $n\to\infty$ this can also represent the drift. Can the authors provide more details on this part? I don't think either claims are clear at all, and I would like to understand the arguments behind this critical step.

With respect to the marginal law, I also wanted to ask a few questions.

9. Are the authors estimating the marginal law $P_t$ at each time of $t$? Does this imply that if the authors were to increase the number of time steps, this would require more estimates?

10. Can the authors provide more details about how $P_t$ is being estimated? I can't seem to find anywhere the authors described the procedure to modelling this density at all.

On a high level, I also have questions regarding the empirical measurements of errors.

11. While MSE going to zero of course implies the method is correctly modelling the underlying dynamics, but it doesn't provide a relative scale of how the methods are performing. Can the authors provide a measure in terms of the Kolmogorov--Smirnov distance, i.e. $L^\infty$ norm of the empirical CDFs between true $p_t$ and estimations?

12. Can the authors also demonstrate how the method improves as a function of computational power, e.g. in terms of the width of the network and maybe the number of particles for the empirical measures case?

---

> ### Author Rebuttal · Authors · 2023-08-09
>
> We thank the reviewer for the detailed comments, questions, suggestions for improvement, and the time spent reviewing the paper. We respond to individual points below.
>
> 1. (Linear factorization of the drift)
>
> We added a discussion of this in the response to all reviewers.
>
> 2. (Different forms of $\varphi$)
>
> We primarily use the form $\varphi(x-y)$ to describe some properties of MV-SDEs. MV-SDEs are often written in this convolutional form and its properties are well studied.
> We use the general form $\varphi(x,y)$ in the proposed architectures to express more general interactions.
>
> The reviewer is correct, we described the MLP with $\varphi(x-y)$. Extending the MLP to support arbitrary interactions is difficult since it requires an operation that repeats the weight for each input point.
> We will note the role of the convolutional form in the description of the MLP architecture.
>
> 3. (When is $\varphi$ known)
>
> The reviewer is correct, if $\varphi$ is known, then we can directly simulate the particle system.
> In practice, it is not clear when $\varphi$ is known, significant domain knowledge is needed.
> A different strategy could be to constrain $\varphi$ to a known class of functions.
> In the well-studied case where $\varphi$ is the gradient of a convex function, we can constrain the parameterization appropriately.
> This would imply aggregation properties for the limiting particle distribution.
>
> 4. (MLP representing MV-SDEs)
>
> These are great points, and we apologize for the confusion.
> To gain some intuition, we first consider the 1 dimensional case where the input $x$, weight matrix $W$ and bias matrix $b$ are of sizes $1$, $K\times 1$ and $K\times 1$. We wish to obtain $K$ repeats of the input particle to interact with the $K$ particles represented by the bias. The weight is thus $K$ repeats of $1$ and the number of rows (or the width) $K$ can be arbitrarily large.
> As $K\to\infty$, we obtain the expectation with respect to the true measure given as the values of the bias.
>
> In the $d$ dimensional case, $x, W, b$ are of sizes $d, K\times d \times d, K\times d$ and $W$ is $K$ repeats of the $d$ dimensional identity matrix to obtain $K$ repeats of the input particle.
> The expectation is then taken with respect to the $K \times d$ bias. As $K \to \infty$, we obtain the true expectation.
> We will include this more detailed explanation in the derivation of the MLP architecture.
>
> (Mean-field layer)
>
> We apologize for the confusion and added a new exposition to this section in the general response to all reviewers.
> We answer specific questions below.
>
> 5. (Distribution of $P_t$)
>
> The distribution $P_t$ defines the particle distribution at time $t$.
> Since each time marginal is absolutely continuous with respect to the Lebesgue measure, the Radon-Nikodym derivative exists.
> That way, if we are approximating a drift of the form: $\mathbb{E}[ \varphi(x, y) ]$ with the expectation with respect to $P_t$, we can rewrite as $\mathbb{E}_{P_0}[\varphi(x, y)\frac{dP_t}{dP_0}]$.
>
> 6. (Radon-Nikodym derivative)
>
> The Radon-Nikodym derivative is a function of $W_0$. The change of measure $\frac{dP_t}{dP_0}$ is applied to the base measure $P_0$ given by the weight matrix $W_0$.
> This leads to an interpretation of the IM architecture as shared weights, re-weighted by the Radon-Nikodym derivative at each time.
>
> 7. (Interpretation of $W_0^{(i)}$)
>
> Exactly, $W_0^{(i)}$ can be thought of as a particle from some distribution that is shared across all time marginals through the change of measure.
>
> 8. (Learning the Radon-Nikodym derivative)
>
> We apologize for the issues in clarity.
> We recall the goal to compute $\mathbb{E}_{y \sim P_t}[\varphi(x, y)]$.
>
> If we consider this to be an empirical expectation, then we can rewrite it as $\frac1n \sum_{i=1}^n\varphi(x, y^{(i)})$ where $y^{(i)}$ are observations with distribution $P_t$.
> This is where the concept of width of the mean-field layer comes into play -- as $n\to\infty$ this empirical expectation becomes exact.
> We write it as an expectation with respect to the empirical measure which is a sum of Dirac measures.
>
> Since we only want to compute an expectation at each time, we can rewrite it as an expectation with respect to a change of measure given by the Radon-Nikodym derivative.
> In particular, the factor $\frac{d P_t}{d P_0}$ is approximated by a neural network with inputs $W_0$ and $t$.
> This allows us to take the expectation with respect to the base measure defined by $W_0$ with a biasing term given by $\frac{d P_t}{d P_0}$.
>
> 9-10. (ML architecture and $P_t$)
>
> We are sorry for the confusion, the marginal law is jointly estimated and penalized to be self-consistent via equations (10) and (11), which are repeated at all time intervals.
> The estimation procedure is described in Algorithm 3.
> In the implementation, we represent $P_t$ as a conditional normalizing flow (the GLOW model) where the conditioning variable is $t$.
> The architecture is also described in Appendix C.3.
>
> 11. (KS Statistic)
>
> This is a great point, we included additional results in the PDF on the KS statistic for the one-dimensional datasets.
>
> 12. (Performance as a function of width and particles)
>
> We thank the reviewer for this suggestion, we included ablation studies in the PDF in the general response.
> For some equations (e.g. the ones requiring jumps) we note empirically that when the width is increased, accuracy is improved until a saturation point.
> Others (e.g. Kuramoto), the width parameter does not play as big of a role, but mainly maintaining the structure of the network improves the performance relative to other methods as shown in Appendix C.4.1.
> We also included figures illustrating the improved convergence rate of the larger width architectures when $\varphi$ is known and unknown.
> We suspect that this is due to favorable properties of the optimization landscape when including more parameters.
> The EM architecture also improves with increased particles as expected.

---

> > ### Comment · Reviewer_DRq5 · 2023-08-12
> > **Response**
> >
> > Thank you for the reply and the added KS statistics. Some of my questions are addressed, but I would like to follow up with the others.
> >
> > On questions 1,2,4, I think my main concern regarding these are some stylistic, but I believe this is important. In particular, I would like to read a paper defining the context of the problems that it's solving clearly, and do not over claim anything that is not exactly matching the claims upon clarification. For example, I don't think it's fair to claim that MLPs can model MV dynamics, as the connection is quite weak in my opinion.
> >
> > W.r.t question 5, is there a difference between $\mathbb{P}_t$ and $P_t$? Also when you say the distribution of the particles at time t, do you mean the joint law over all the particles?
> >
> > For question 8, can you clarify exactly how you get the neural network to represent and learn this Randon--Nikodym derivative? Like I would like to understand the setup here, I'm genuinely not sure where the signal is coming from, and what the loss is etc.
> >
> > So I know I asked for the KS statistic in the review only, so you didn't have much time to experiment further, but I am somewhat concerned the KS statistics are quite large. Since the CDFs are functions contained in $[0,1]$, I would hope the KS statistics are less than 0.1 by a successful method. A large KS statistic can mean many things, but most likely the numerical method is not quite capturing the same distribution. You can plot the two empirical CDFs, or histograms/kernel density estimates, so that you can visually examine them and that should paint a clear picture.
> >
> > Honestly speaking, at this point I don't think I'm too convinced yet, but feel free to respond further so we can continue the discussion.

---

> > > ### Author Response · Authors · 2023-08-14
> > >
> > > We appreciate the follow up and would also like to thank the reviewer for the opportunity to discuss.
> > >
> > > 1. Please allow us to clarify our claims on neural architectures, only two proposed neural architectures for representing MV-SDEs, implicit measure (IM) and marginal law (ML) architectures, based on learned measures and generative networks (line 71).
> > > These architectures are able to represent the general form of MV-SDEs, including general factorization of the drift and general form of interaction.
> > > We will introduce the MLP architecture only for motivation, relegate the bulk of the MLP exposition to the appendix, and clarify in the main text the assumptions and weak connection of MLPs to MV-SDEs.
> > >
> > > For consistency, we begin by clearly defining the context of the problem we are solving as modeling and inferring parameters for MV-SDEs with linearly factorized drift and general interaction:
> > > $$b(X_t,p_t,t)=f(X_t,t)+E_{y\sim p_t}[\varphi(X_t,y)]$$
> > > where $p_t=\mathrm{Law}(X_t)$. The particles $X_t$ are exchangeable and distributed as $p_t$.
> > >
> > > Then, to prevent misunderstandings, we restate our contributions on neural architectures as only the implicit measure (IM) and marginal law (ML) architectures, with the mean-field components summarized as:
> > > $$\mathrm{IM}(X_t): E_{y\sim p_t}[\varphi(X_t,y)]=E_{y\sim p_0}[\varphi(X_t,y)\frac{\mathrm{d}P_t}{\mathrm{d}P_0}]\approx\frac{1}{K}\sum_{i=1}^K[\varphi(X_t,W_0^{(i)},t;\theta)].$$
> > > $$\mathrm{ML}(X_t): E_{y\sim p_t}[\varphi(X_t,y)]\approx\frac{1}{K}\sum_{i=1}^K[\varphi(X_t,Y_t^{(i)};\theta)], \quad Y_t^{(i)} \sim p(\varepsilon^{(i)},t;\phi).$$
> > > where in the IM architecture, $\varphi(\cdot,W_0,t;\theta)$ is a MLP with inputs $X_t, W_0$ and $t$ that approximates the combination of the interaction function $\varphi$ with inputs $X_t$ and $y_t$, and the change of measure with inputs $y_0$, represented by weight $W_0$, and $t$;
> > > and in the ML architecture $\varphi(\cdot, \cdot;\theta)$ is a MLP with inputs $X_t, Y_t$, and $p(\cdot, t;\phi)$ is a generative architecture with an input noise source $\varepsilon$ and conditioning on time $t$.
> > >
> > > In addition, to prevent over claims, we will emphasize the weak connection of MLPs to MV-SDEs, specifically MV-SDEs with the linearly factorized drift and convolutional interaction, introduce it only for motivating the representation of expectations with MLPs and relegate the bulk of the MLP exposition to the appendix.
> > >
> > > With regards to questions 1, 2, 4:
> > >
> > > a. We motivate with the form of the linearly factored drift but this condition is not necessary in the proposed methods.
> > >
> > > b. We consider the convolutional form of the MV process due to its ubiquity in the literature. We only require this structure in the MLP architecture and we do not impose this structure in the other architectures.
> > >
> > > c. We note the MLP approximation of the MV process holds in the case where the convolutional form is given. The MLP is only used as motivation and is not used as a main contribution of the work.
> > >
> > > It is not our intention to over claim, please let us know what else the reviewer believes is oversold and we will make adjustments to the final manuscript.
> > >
> > > 2. $P_t$ and $\mathbb{P}_t$ are the same, and yes this refers to the joint law of all particles at time $t$. Apologies for the confusion here, we originally used $\mathbb{P}_t$ in the Radon-Nikodym derivative since that is conventionally written using the blackboard font but realized it may introduce more confusion when switching to referring to the density only.
> > > We were aiming for a consistent notation and will use $p_t$ and $\mathrm{d} P_t$ in the final manuscript.
> > >
> > > 3. The Radon-Nikodym derivative can be understood as a positive function that takes as inputs a point $y$ and time $t$ and outputs the weight of that point at time $t$ such that the weighted expectation matches the true expectation.
> > > To parameterize such an object, we need to represent a function $\lambda$ that maps $y, t \to \mathbb{R}^+$.
> > > This can be done using a neural network.
> > > Putting this together with the interaction function $\varphi$, we have
> > > $$
> > > \mathrm{IM}(X_t) := \frac1K \sum_{i=1}^K\varphi(X_t, W_0^{(i)}; \theta) \lambda(W_0^{(i)}, t; \theta) = \frac1K \sum_{i=1}^K \varphi(X_t, W_0^{(i)}, t; \theta)
> > > $$
> > > where $\lambda$ represents the weight on each hypothetical particle of the base measure and we can combine the product $\varphi(X_t, W_0^{(i)};\theta)\lambda(W_0^{(i)},t;\theta)$ into a single term $\varphi(X_t, W_0^{(i)}, t;\theta)$ represented with a single MLP.
> > >
> > > For estimation, we perform maximum likelihood estimation (MLE) with the likelihood given in equation (8) derived from Girsanov's theorem that takes as input the modeled drift and the observations. The estimation procedure is general and works across the proposed architectures. We thus use the IM parameterization and perform the same estimation techniques that we proposed (MLE) in the rest of the text. If the drift is correct, an appropriate Radon-Nikodym derivative was learned.

---

> > > > ### Author Response · Authors · 2023-08-14
> > > > **Discussion (continued)**
> > > >
> > > > 4. The accuracy at which the distribution will be estimated is a function of the number of sample points available.
> > > > If we use more points for estimation, all the methods improve on their KS statistic.
> > > > Additionally, the empirical CDFs (ECDFs) tend to match, but since the KS statistic is the supremum over the difference between the CDFs, this tends to be a stronger metric.
> > > > Instead of the max, we also consider the mean, the 75th and 90th percentile in the following table with 100 particles with noise 0.1 for the mean-field atlas model:
> > > >
> > > >
> > > > |     | mean ECDF dist | 75\% ECDF dist | 90\% ECDF dist | KS          |
> > > > |-----|----------------|----------------|----------------|-------------|
> > > > | MLP |  0.09 (0.013)  | 0.16 (0.017)   | 0.25 (0.017)   | 0.31 (0.019)|
> > > > | IM  |  0.02 (0.004)  | 0.02 (0.006)   | 0.05 (0.012)   | 0.10 (0.010)  |
> > > > | ML  |  0.02 (0.004)  | 0.03 (0.005)   | 0.07 (0.017)   | 0.14 (0.025)|
> > > > | EM  |  0.02 (0.001)  | 0.02 (0.001)   | 0.05 (0.004)   | 0.10 (0.006) |
> > > >
> > > > In this case, the results for the EM architecture with 100 training particles represent some sort of lower `bound' on the estimation accuracy. Ideally we could derive a lower bound on the estimation accuracy as a function of the number of samples but we leave that as future work.
> > > >
> > > >
> > > > For the other mean-field atlas experiments in the rebuttal PDF using this format:
> > > >
> > > >
> > > > |             | mean ECDF dist | 75\% ECDF dist | 90\% ECDF dist | KS          |
> > > > |-------------|----------------|----------------|----------------|-------------|
> > > > | Noise = 1.0 |                |                |                |
> > > > | MLP         | 0.1 (0.01)     | 0.16 (0.03)    | 0.27 (0.03)    | 0.35 (0.03) |
> > > > | IM          | 0.05 (0.01)    | 0.07 (0.02)    | 0.13 (0.03)    | 0.21 (0.03) |
> > > > | ML          | 0.05 (0.02)    | 0.07 (0.03)    | 0.14 (0.05)    | 0.24 (0.03) |
> > > > | EM          | 0.05 (0.01)    | 0.07 (0.03)    | 0.15 (0.04)    | 0.24 (0.03) |
> > > > | Noise = 0.5 |                |                |                |
> > > > | MLP         | 0.09 (0.01)    | 0.16 (0.02)    | 0.26 (0.02)    | 0.34 (0.03) |
> > > > | IM          | 0.03 (0.01)    | 0.04 (0.01)    | 0.08 (0.02)    | 0.15 (0.02) |
> > > > | ML          | 0.04 (0.01)    | 0.06 (0.02)    | 0.12 (0.03)    | 0.21 (0.03) |
> > > > | EM          | 0.03 (0.01)    | 0.04 (0.01)    | 0.07 (0.02)    | 0.14 (0.02) |
> > > > | Noise = 0.1 |                |                |                |
> > > > | MLP         | 0.1 (0.01)     | 0.16 (0.03)    | 0.26 (0.03)    | 0.34 (0.02) |
> > > > | IM          | 0.03 (0.01)    | 0.04 (0.01)    | 0.08 (0.02)    | 0.14 (0.02) |
> > > > | ML          | 0.04 (0.01)    | 0.06 (0.02)    | 0.13 (0.03)    | 0.21 (0.02) |
> > > > | EM          | 0.03 (0.01)    | 0.04 (0.01)    | 0.08 (0.02)    | 0.14 (0.02) |
> > > >
> > > >
> > > > We hope that these notes help paint a clear picture. Please let us know any other points that need clarification.

---

> > > > > ### Comment · Reviewer_DRq5 · 2023-08-16
> > > > > **Response**
> > > > >
> > > > > Thank you for the lengthy and detailed response. Given that I have a fairly large reviewing load (and other urgent tasks outside of reviewing), I want to just focus on this point of KS statistic for the rest of this discussion.
> > > > >
> > > > > Perhaps let me clarify my intention further. I am hoping to see a method that effectively serves as a discretization method for MV-SDEs. Typically in numerical methods, an important measure of success is the rate of which the errors improve as a function of computing resources spent. Furthermore, one can usually use a log-log plot to see the exact rate of which it converges, and the errors being measured should be quite small (visually almost negligible) for the best numerical solution.
> > > > >
> > > > > While I tend to agree that KS is relatively strong metric, which is why I'm not expecting to see values on the scale of 0.01, but rather just under 0.1 should be a sufficient demonstration. I see that some of the new KS statistics reported are near this scale I am hoping for. Can the authors demonstrate that with even more compute, we can indeed get to arbitrarily small KS, and hence demonstrate convergence? This was essentially what I'm hoping for.
> > > > >
> > > > > Given there's not a lot of time left in the discussion period, perhaps let's just focus on the easiest example, and no need to reproduce experiments for all of them. Once again, I will raise my score if the authors convince me on this point.

---

> > > > > > ### Author Response · Authors · 2023-08-17
> > > > > >
> > > > > > We very much appreciate the reviewer circling back to us.
> > > > > >
> > > > > > Focusing on just the point of the KS statistic, indeed, with more compute, we can get to arbitrarily small KS and thus demonstrate convergence.
> > > > > > Here we include an additional study on the IM architecture for the 1-d Kuramoto problem with no observation noise and show that the expected behavior occurs:
> > > > > >
> > > > > > This is KS averaged over time marginals.
> > > > > > | N   | KS (IM)         |
> > > > > > |-----|-----------------|
> > > > > > | 10  |  0.1222 (0.041) |
> > > > > > | 50  |  0.0792 (0.038) |
> > > > > > | 100 |  0.0544 (0.008) |
> > > > > >
> > > > > > This is KS over all time marginals together.
> > > > > > | N   | KS (IM)         |
> > > > > > |-----|-----------------|
> > > > > > | 10  |  0.0893 (0.043)|
> > > > > > | 50  |  0.0481 (0.044)|
> > > > > > | 100 |  0.0268 (0.015)|
> > > > > >
> > > > > > We hope that this clears up any confusion regarding how the estimator scales with an increased computational budget.
> > > > > >
> > > > > > We thank you again for your thoughtful comments.

---

> > > > > > > ### Author Response · Authors · 2023-08-20
> > > > > > > **Any Other way that we can be of assistance?**
> > > > > > >
> > > > > > > Once again we thank the reviewer very much. We truly appreciate the reviewer's feedback.
> > > > > > >
> > > > > > > Since the correspondence period is soon coming to end, we write to gently ask if (besides the simulation that your requested and we provided above) there is any other way that we can be of assistance?
> > > > > > >
> > > > > > > Again thanks a million for all your comments and efforts in spite of your busy schedule.

---

### Official Review · Reviewer_KoGe · 2023-07-23

**Soundness:** 4 excellent
**Presentation:** 4 excellent
**Contribution:** 3 good
**Rating:** 7
**Confidence:** 2

**Summary:**

This paper proposes a methodology for simulating McKean-Vlasov (mean-field) equations using standard function approximation techniques, e.g. neural networks. It provides mathematical intuition for these algorithms and evaluates them on a broad suite of benchmarks.

**Strengths:**

The paper is very well written with clear exposition of its main points. The proofs of key claims seem broadly correct as well.

The methodology is broadly well justified and uses very intuitive ideas from stochastic analysis.

In particular, the adaptation of standard neural network techniques for simulating ODEs/SDEs is not entirely applicable here, and so the derived techniques need to account for the estimation of the particle density. The resulting algorithm is novel and an important independent contribution.

The experiment evidence, especially in the Gaussian case, seems to vindicate the intuition of this algorithm and clearly outperforms the chosen baselines.

**Weaknesses:**

I would say that the ultimate idea is rather simple, i.e. approximating both the drift function (both interactive and interaction-free), and possibly the particle density with some kind of learned approximations.

I appreciate the inclusion of standard deviations in the Tables, however these values seem, particularly in Table 1, to be quite large relative to the proposed gains.

I have some additional questions about the methodology. To summarize, I think this paper makes fairly solid contributions to the simulation of McKean-Vlasov equations, which is an important problem. If my questions are addressed, I would be amenable to raising my score.

**Questions:**

How is the Radon-Nikodym change-of-measure learnt in the implicit model? This seems like a difficult task and I would be skeptical that it would be done without large amounts of reference data. Consequently, I am not sure what the clear advantage of IM over EM would be. Could the authors elaborate?

The usage of sample trajectories from Eq. (12) in the objective (11) seems quite non-trivial if the constraint in (11) is exact. In particular, I don’t see how (11) could be easily enforced. It seems that this is not being done exactly from Algorithms 3 and 4 in the appendix, so this should probably be clarified in the main text.

I would argue that Methods 3.2 and 3.3 are more similar than given in Figure 3, since both are essentially proposing to learn the densities (but one using a Radom-Nikodym w.r.t. Time $0$ and the other a generative model). However the emphasis on neural networks in the implicit method is quite important, so perhaps the current organization is OK.

The first line in sentence 180 is redundant given the definitions.

Remark 3.2 is probably better cited from Villani, particularly Section 8.3 of Topics or 9.6 of Topics.

**Typos:**
L. 591 -> I don’t think this should be pointing to section C.2.3, but rather to C.3 or something like that.

**References**
Villani, C., 2021. Topics in optimal transportation (Vol. 58). American Mathematical Soc..

---

> ### Author Rebuttal · Authors · 2023-08-09
>
> We would like to thank the reviewer for the well-thought feedback and comments.
> We address these individually below.
>
> - (Simplicity in the ideas)
>
> We hope that in addition to the methodological contributions, the motivations behind studying distributional dependence within the context of stochastic processes in machine learning is received as a new and interesting approach.
> In addition, we showed with some preliminary results that compared to a generic MLP, using the MF layer and generative model brought about some theoretically and empirically justified improvements, such as the implicit regularization, relation to attention and the ability to model jumps without a jump noise process.
>
> - (Large variances)
>
> The reviewer is correct, the performances of the proposed methods in the real datasets are difficult to compare since they tend to have higher variances.
> We note that for the probabilistic modeling experiments, we mainly wanted to highlight the performance compared to the MLP architecture, which in many of the scenarios the proposed methods show an improvement.
> To add additional context, we included other common density estimation methods, but we are mainly testing whether the architectures explicitly informed by the McKean-Vlasov structure can improve under the same estimation technique.
> For the synthetic experiments, the differences between the architectures become more apparent.
> We will highlight this within the discussion of the results and the limitations.
>
>
> 1.a. (How the change of measure is estimated)
>
> We apologize for the confusion regarding this computation.
> In the implicit model, the base measure $P_0$ is given by the weight $W_0$, the change of measure $\frac{d P_t}{d P_0}$ is approximated by a neural network with inputs $W_0$ and time $t$.
> The change of measure is estimated jointly with the network parameters through the ELBO.
> We will rewrite according to the note on the mean-field layer in the general response.
>
>
> 1.b. (Advantage of IM over EM architecture)
>
> For the EM architecture, the inputs to the network need to be the $N$ sample points that we observe at each time in order to compute the empirical expectation of the $N$ particles influencing the current particle.
> For all time marginals, there is no learned measure. We need the population of $N$ particles, thus the name empirical measure.
> If there are too few particles, the empirical expectation may have high variance. We added an ablation study on the number of particles in the PDF in the general response.
>
> The IM architecture on the other hand represents the time marginal law $P_t$ implicitly through the base measure $P_0$ and the change of measure $\frac{d P_t}{d P_0}$. To compute the expectation, we do not need a different set of samples at each time point.
> Specifically, the factor $\frac{d P_t}{d P_0}$ acts as an importance sampling weight applied to the base particles of $W_0$ which is shared across time.
> It also acts like an interpolator in cases where there are too few data points.
> In addition, if we consider the case of the stationary measure (i.e. $P_t \to P_\infty$), then the IM architecture needs to only estimate the best fitting weight $W_0$.
> This is why we studied the implicit regularization of the IM architecture in section 5 so that we can better understand this interpolating behavior.
>
> 2. (Enforcing the constraint)
>
> The reviewer is correct, the constraint is not exactly satisfied but included as a penalty term in the loss function during optimization.
> This is done using sampling, where we sample the trajectory and compare the expectation of the samples versus the function itself.
> We briefly described the procedure in Algorithm 3, and will further clarify this in the final manuscript.
>
> 3. (Differences between methods 3.2 and 3.3)
>
> The reviewer is correct, both approaches model distributional dependence, one with a base measure and change of measure and the other with a generative model.
> The main differences between the IM and ML architectures is the IM architecture requires integrating according to a sampling scheme like the Euler-Maruyama method to obtain marginal observations while the ML architecture allows sampling of arbitrary time marginals.
> In that sense, we have access to the full density at each time marginal with the ML architecture.
>
> 4. (Redundancy in line 180)
>
> We thank the reviewer for pointing that out, we will remove this line or rewrite it in a way such that we recall the properties of the original assumptions.
>
> 5. (Citation)
>
> This is a great point, we included the citation in Remark 5.2 originally to Santambrogio's manuscript due to its ease in accessibility but we will modify the citation to Villani's book.
>
> 6. (Incorrect hyperlink)
>
> We thank the reviewer for pointing that out, we will correct the link to Appendix C.3 in the final manuscript.

---

> > ### Comment · Reviewer_KoGe · 2023-08-20
> > **Response**
> >
> > I thank the authors for their prompt and comprehensive response. I agree in principle that this paper makes some important contributions in terms of building a practical framework for simulating MV-SDEs. Having reviewed both the response to my initial remarks and those of the other reviewers, it seems that the presentation of the paper could yet be improved in many aspects.
> >
> > As the authors have addressed my main concerns (in particular I appreciate the additional KS statistics in the Response to Review DRq5, which are quite convincing), I will raise my score to a 7, contingent on improvements being made to the presentation in the final draft.

---

> > > ### Author Response · Authors · 2023-08-20
> > > **Thank you**
> > >
> > > Again, Thank you very much for your detailed comments which will significantly improve the final version of our paper. We will incorporate all the comments that you have made.
> > >
> > > Again, a million thanks for your thoughtful review and comments.

---

### Author Rebuttal · Authors · 2023-08-09

We would like to thank the reviewers for their helpful feedback and all the comments in helping improve the paper. Here we include the common responses to all reviewers:

### Mean-field layer exposition

Let us first define the measure of the particles at time $t$ to be $P_t$ and an arbitrary base measure of the particles at a time (call it `$0$') as $P_0$.
Our goal for this architecture is to
a) represent the base measure $P_0$ as a sum of Dirac functions; and,
b) change the measure for each time such that the expectation is correct for each time, i.e. $\mathbb{E}[\varphi(x, y) ]=\mathbb{E}_{P_0}[\varphi(x, y) \frac{dP_t}{dP_0}]$.
Both the base measure $P_0$ and the Radon-Nikodym derivative $\frac{dP_t}{dP_0}$ are modeled and learnt.

Before we describe how this is done, we want to note the motivations behind this representation.
We no longer need particles at all times $t$ -- rather, we just need to know the base measure and change of measure.
Furthermore, we can use this concept to interpolate between time points using the change of measure that is a function of the base measure and time.

To represent the base measure $P_0$, we consider a weight matrix $W_0 \in \mathbb{R}^{K \times d}$ with the number of rows (or the width) equal to $K$. Each row of the weight matrix may be seen as a hypothetical particle of dimension $d$.
To represent the change of measure $\frac{dP_t}{dP_0}$, we consider a neural network with inputs $W_0$ and $t$ to approximate $\frac{dP_t}{dP_0}$ for all $t$.
This is formalized by the mean-field layer which includes both of these components, $P_0$ and $\frac{dP_t}{dP_0}$.
We can think of this as a particular type of neural network where the weight $W_0$ representing $P_0$ is shared across different time steps through a re-weighting factor $\frac{dP_t}{dP_0}$.
The key is that by including the mean-field layer, we can easily and explicitly represent complex interactions between each input $X_t$ and the measure $P_t$.


### Important models with linearly factorized drift

The drift can be factored linearly in a number of models that have been studied in the literature, we will list a few below:
- The mean-field Fitzhugh-Nagumo equation which models neural spikes.
- The Kuramoto equation which models oscillators.
- The opinion dynamics model which models the interactions of opinions of individuals.
These equations also appear in Appendix C.2.1 where we described the synthetic experiments.

We note, however, that the linear decomposition of the drift is not necessary to use the proposed architectures, but we present it this way for ease of exposition, and since many important models in the literature assume this form.
In implementation, the proposed architectures support the form of $g(\mathbb{E}[ \varphi(x,y)], x)$ for representing the drift, allowing application to more general scenarios without linearly factorized drift, therefore no additional conditions are required.
We will motivate the linear factorization and also note the general representation in the final manuscript.


#### Tables and figures in the PDF

We include the following tables in the PDF:

1,2. The Kolmogorov–Smirnov statistic for experiments that are 1-dimensional.

3. Ablations on the accuracy of IM architecture with different widths.

4. Ablations on the accuracy of EM architecture given different number of observations.

We include the following figures in the PDF:

1,2. Ablations on the convergence of IM architecture with different widths.

---

### Decision · Program_Chairs · 2023-09-21

**Decision:**

Reject

**Comment:**

This paper is reviewed by 4 expert reviewers and received the following Rating/Confidence scores: 8/3, 2/4, 7/2, 6/3.

- The reviewer (2/4) is particularly critical and even after the discussion period, their concern on on the validity of the task "predicting the jumping behavior in observations via learning the interaction kernel of the MV-SDE" remains. Based on the reviewer's experience, they expect that the neural network is able to learn the interaction kernel if it is sufficiently regular. However, given a regular interaction kernel, the reviewer thinks it is possible for the trajectory to have jumping behavior within a finite evolving time. While the authors did provide an example, the reviewer was not able to verify if the interaction kernel to be learned is even continuous.

- The reviewer (6/3) increase their score but added in the AC/reviewer discussion period that they reserved some of their thoughts from the review, mainly that they found the paper relatively difficult to read, with sloppy notations and missing definitions, somewhat misleading in places, although the authors seem to claim these are addressable issues.The reviewer also pointed that the long back and forth is also wearing them down. Even if they ended up increasing their score, they reduced their confidence.

Given the above arguments, this paper is at the borderline. After reading the paper and the reviews, I partially agree with the reviewers and I found that the main issue with the paper is its clarity and the presentation. I believe that many of the issues raised by the reviewers can be addressed but this will require significant revision on the paper. Therefore, AC recommends not including this paper to the program and the authors should carefully address the concerns raised in the next version of their paper.